# Feature Correspondences Increase and Hybrid Terms Optimization Warp for Image Stitching

**DOI:** 10.3390/e25010106

**Published:** 2023-01-04

**Authors:** Yizhi Cong, Yan Wang, Wenju Hou, Wei Pang

**Affiliations:** 1Key Laboratory of Symbolic Computation and Knowledge Engineering of Ministry of Education, School of Artificial Intelligence, Jilin University, Changchun 130012, China; 2College of Computer Science and Technology, Jilin University, Changchun 130012, China; 3School of Mathematical and Computer Sciences, Heriot-Watt University, Edinburgh EH14 4AS, UK

**Keywords:** image stitching, image alignment, feature correspondences increase, hybrid terms optimization

## Abstract

Feature detection and correct matching are the basis of the image stitching process. Whether the matching is correct and the number of matches directly affect the quality of the final stitching results. At present, almost all image stitching methods use SIFT+RANSAC pattern to extract and match feature points. However, it is difficult to obtain sufficient correct matching points in low-textured or repetitively-textured regions, resulting in insufficient matching points in the overlapping region, and this further leads to the warping model being estimated erroneously. In this paper, we propose a novel and flexible approach by increasing feature correspondences and optimizing hybrid terms. It can obtain sufficient correct feature correspondences in the overlapping region with low-textured or repetitively-textured areas to eliminate misalignment. When a weak texture and large parallax coexist in the overlapping region, the alignment and distortion often restrict each other and are difficult to balance. Accurate alignment is often accompanied by projection distortion and perspective distortion. Regarding this, we propose hybrid terms optimization warp, which combines global similarity transformations on the basis of initial global homography and estimates the optimal warping by adjusting various term parameters. By doing this, we can mitigate projection distortion and perspective distortion, while effectively balancing alignment and distortion. The experimental results demonstrate that the proposed method outperforms the state-of-the-art in accurate alignment on images with low-textured areas in the overlapping region, and the stitching results have less perspective and projection distortion.

## 1. Introduction

Image stitching is the process of merging a series of images with overlapping regions into a high-resolution image with a wider field of view [1]. It has been widely used in applications such as panorama [2], autonomous driving [3], video stabilization [4], virtual reality [5], and many others. Up till now, there are still great challenges for image stitching to align accurately, eliminate artifacts, and mitigate distortion, especially on low-textured and large-parallax images.

Traditional image stitching methods often use feature points to obtain a global homography matrix, and they map images with overlapping regions to the same plane. However, these approaches only work if the images are in the same plane or only rotated around the camera center. Under the condition of both parallax and rotation, global homography often produces misalignment and/or artifacts in the overlapping region and distortion in the non-overlapping region. Recently, spatially varying warping [6,7,8] has been widely used in image stitching. It divides an image into several grid cells, and then aligns each grid cell with local homography. It can effectively eliminate misalignment, but it may also produce distortions in the non-overlapping region. Since homography-based methods inevitably generate distortions, researchers have taken a different approach by using a seam-driven strategy [9,10,11], which finds the optimal seam according to a customized energy function. Although such an approach can effectively mitigate distortions, it usually cannot achieve the desired alignments. In addition, such a method relies too much on the energy function, and it is more frequently used as a post-processing blending method to eliminate artifacts. In recent years, deep learning-based image stitching frameworks [12,13,14] have been widely studied by researchers. However, they may still not be able to achieve the desired accurate alignment and distortion preservation performance compared to traditional methods.

Feature detection and correct matching are the basis of the image stitching process. SIFT [15] is a feature point detection and matching method used by almost all image stitching methods. After preliminary matching using SIFT, RANSA C [16] is used to remove outliers and obtain inliers. Although SIFT can extract a large number of feature points, the correct matching points after RANSAC iteration are greatly reduced compared with the detected feature points, which may lead to insufficient feature correspondence in the low-textured or repetitively-textured regions. Figure 1 shows the feature correspondences of SIFT+RANSAC under three different image types. It can be seen that the traditional SIFT+RANSAC pattern has sufficient inliers only under the condition of dense texture in the overlapping region, while obviously insufficient inliers in low-textured areas (see Figure 1a,c).

Grid-based image stitching methods are heavily dependent on the quantity and quality of feature points, which are crucial to the quality of image stitching results. The as-projective-as-possible (APAP) warp model [8] is a typical mesh-based local warping model. It uses SIFT+RANSAC as pre-processing. Figure 2a shows the inliers of APAP and the stitching result. The input images are with low-textured or repetitively-textured regions, which lead to insufficient inliers after RANSAC. It further causes misalignment of the stitching result. RANSAC can increase the number of inliers by increasing the threshold. Figure 2b shows the inliers and the stitching result after increasing the RANSAC threshold appropriately. Obviously, increasing the number of feature correspondences produces a more accurate alignment. However, limited by the local alignment of the APAP method, some unacceptable jagged misalignments appear unnaturally. Furthermore, limited by the characteristics of the RANSAC algorithm itself, when the RANSAC threshold is too high, it may still generate false matches if the overlapping region has many similar textures.

Considering that there may still be the issue of matching errors after RANSAC, elastic local alignment (ELA) [19] introduces Bayesian feature refinement to further remove possible outliers. Although outliers hardly exist after using ELA, ELA also removes some of the original normal values, and this may lead to the issue of insufficient feature points. Leveraging line-point consistence (LPC) [20] establishes a relationship between point and line dual features, and it applies a characteristic number (CN) [21] to add salient points to the feature point set to enrich the number of feature points. However, the feature points added by CN may not be correct matches, and they need to go through RANSAC, which will also make a low-textured region with insufficient feature points (see Figure 2c). Although some methods [18,22,23] that introduce line features and use LSD [24] to extract line segment features can cope well with a low-textured environment, line segments are mainly matched by endpoints, which are essentially point features, and they are subject to line segment length thresholds. Currently, line segment matching is still a challenging problem.

Grid-based Motion Statistics (GMS) [25] assume that adjacent pixels share similar motions, which is consistent with the principle of image stitching. It uses ORB [26] to extract feature points. Since ORB can set the initial feature point threshold, combined with GMS, it can increase the number of correct matching points in low-textured regions, thereby ensuring that the warped model has sufficient anchor points for local alignment. Compared with the SIFT+RANSAC pattern, the ORB+GMS method can not only make feature correspondences sufficient, but also ensure the correctness of the feature matching, which helps improve the quality of the stitching. The results of the proposed method are shown in Figure 2d.

In this paper, we propose a novel image stitching method based on feature correspondences increase and hybrid terms optimization warp. We first use the ORB+GMS algorithm to extract and match feature points to obtain sufficient feature correspondences, and at the same time we use line features to assist alignment and structure preservation; then, an inital homography combining global homography transformation and global similarity transformation is estimated as the initial homography to be optimized. On this basis, we estimate various optimization terms for the final image warping. Finally, we obtain the final stitching result by linear blending or seam-cutting blending. The overall process architecture of our method is shown in Figure 3.

The contributions of our approach are as follows:A novel method to increase feature correspondences is proposed, which can be added to any image stitching model. It solves the problem of insufficient feature correspondence in low-textured or repetitively-textured regions, and it can effectively eliminate misalignment and artifacts.A novel hybrid transformation combining global homography transformation and global similarity transformation is proposed to serve as the initial homography for structure preservation, which can flexibly fine-tune the structure of image stitching results compared to conventional global homography.Various optimization terms are used to locally adjust the above hybrid warping model, which can effectively mitigate projection and perspective distortion. Our flexible and robust method can effectively balance alignment and distortion, especially on images with low-textured areas in the overlapping region. For images with large parallax or significant foreground-background relationship, seam-cutting blending instead of linear blending is used to eliminate inevitable misalignment or artifacts.

## 2. Related Work

### 2.1. Feature Extraction and Matching

Feature points refer to the points where the gray values of the image change drastically or the points with large curvature on the edges of the image, such as corners and spots. In the early days, common corner detection algorithms were Harris corner [27] and FAST [28]. Due to the feature limitations of corner points, Lowe et al. proposed SIFT [15], which is invariant to scale, rotation, and brightness. It is robust and fast, and it has been the most popular feature detector until now. SURF [29] improves the speed compared with SIFT, which is more suitable for practical applications. BRIEF [30] uses the Hamming Distance to achieve matching using the XOR operation between bits. ORB [26] combines the advantages of FAST and BRIEF. Compared with SIFT/SURF, it can detect denser matching points in areas with flat textures, and it has the best comprehensive performance among the current traditional feature detectors.

After feature detection, the Brute Force (BF) [31] algorithm is often used for preliminary matching, but there are often some false matches. In recent years, RANSAC [16] has been the most popular method for filtering out outliers. However, it is affected by the threshold of inliers; especially for locally aligned warping models, inliers are often mistakenly eliminated by RANSAC, resulting in an insufficient number of points in flat textured areas, and the rest cannot be guaranteed to be inliers, resulting in wrong stitching results. Some researchers have taken corresponding countermeasures to the problem of RANSAC. The triangular facet transformation (TFT) [32] uses bundle adjustment (BA) [33] to optimize for false matches that may be due to noise. LPC [21] uses the CN [20] point-line relationship to expand the matching point set, but the two-step RANSAC may lead to the loss of feature correspondences in the weakly-textured region, thus affecting the stitching results. Bian et al. proposed GMS [25], which assumes that adjacent pixels have similar motion and distinguishes between correct and incorrect matches by computing statistical scores within a small area around the matching point. It can be combined with ORB to increase the number of features by increasing the matching point threshold to help the matching quality, which makes it possible to solve challenging image stitching problems. Therefore, compared to RANSAC, GMS is more robust to address feature matching errors and insufficient features in low-texture regions.

### 2.2. Spatially Varying Warping and Seam Cutting

Traditional image stitching uses a global homography [2] to warp the image in the ideal case where the scene depth has little change in overlapping regions. In recent years, researchers have used multi-homography spatially varying warping to deal with parallax. The dual-homography warping (DHW) [34] divides the scene into a background plane and a foreground plane, and it aligns the background and foreground with two homography matrices, respectively. Smoothly varying affine (SVA) [7] improves local transformation and alignment with multiple affine transformations. APAP [8] divides the image into uniform grid cells, and it performs local homography alignment for each grid, which can better cope with parallax scenes. However, grid-based spatially varying warping often produces severe distortion in non-overlapping regions. The adaptive as-natural-as-possible (AANAP) warp [17] combines local homography transformation with global similarity transformation to effectively reduce distortion in non-overlapping regions. The shape-preserving half projective (SPHP) warp [35] uses subregional warping to smoothly transition overlapping and non-overlapping regions to reduce distortion.

Due to the difference in parallax and image capture time, artifacts inevitably appear in the overlapping areas of the stitching results. Gao et al. proposed a seam-driven [9] approach to deal with parallax and eliminate artifacts. Parallax-tolerant Image Stitching [10] is inspired by content-preserving warps (CPW) [36] and optimizes the use of seams to improve the stitching performance of large parallax scenes. Seam-guided local alignment (SEAGULL) [11] adds curved and straight structural protection constraints. Perception-based seam cutting for image stitching [37] introduces human perception and saliency detection to obtain the optimal seam, making the stitching result look more natural. Xue et al. proposed a stable seam measurement [38], which is used to estimate a stable seam cutting for mitigating parallax via a hybrid actor–critic architecture.

### 2.3. Structure Preservation and Distortion Mitigation

Lines are common features in natural images. Line structures bend unnaturally after image warping. The dual-feature warp (DFW) [18] introduces the line segment feature, and the constraint term constrains both points and lines to achieve better alignment. The global similarity prior (GSP) [39] constrains globally similar angles with line alignment, while local and global constraints are combined to reduce structural distortion. The quasi-homography warp (QHW) [40] uses intersecting lines to preserve the nature of homography warping. The single-perspective warp (SPW) [22] utilizes the QHW characteristic; it introduces a variety of protection terms and uses the least square method to obtain the optimal solution of the objective function to obtain the optimal homography. LPC [21] introduces the concept of global lines on the basis of SPW, and it merges short line segments into long ones to reduce the bending of line structures in natural images. Similarity transformation has been proved in [34] to reduce distortion in the overlapping region, and the reason being that similarity transformation are more rectangular and look more natural than homography. Analogously, the work in [41] introduces line and regular boundary preservation constraints to make the boundary of the stitching result as close as possible to a rectangle, while reducing unwanted distortions.

The remainder of this paper is organized as follows. Section 3 presents the proposed image stitching algorithm. The detailed experimental procedure and results are presented in Section 4. Finally, the conclusions are summarized in Section 5.

## 3. The Proposed Method

The homography transformation is the most common method to achieve image stitching. For the single image warp, global homography warping can handle the cases in the ideal conditions. However, under low texture and large parallax conditions, it is still a big challenge to align accurately even with local homography. It is very dependent on the quality and quantity of features correspondences in the pre-processing stage. The traditional SIFT+RANSAC pattern has been unable to satisfy the above requirements. Meanwhile, the homography transformation inevitably introduces distortion in the non-overlapping region. How to balance alignment and distortion has gradually become the key to single image warping. In this section, we first describe our feature correspondences increase module, and then we propose our hybrid terms optimization warp to address the balance of alignment and distortion.

### 3.1. Feature Correspondences Increase

Inspired by GMS, we merge the feature correspondences increase module into the image stitching process to solve the problem of insufficient feature correspondences in low-textured areas in the overlapping region. Compared with the one-step RANSAC, we try to get more correct matches in low-textured or repetitively-textured regions to satisfy the local warping model’s demand for sufficient feature points in overlapping regions to achieve more accurate alignment. Here, we use ORB with high robustness and dense feature points as the detector, BF for preliminary matching, and GMS refinement to distinguish between correct and incorrect matches.

GMS considers that there are several features that match the matching relationship in the correctly matched neighborhoods, while there are almost no features in the incorrectly matched neighborhoods. According to this feature, GMS counts the number of features that match the matching relationship to distinguish between correct and incorrect matches. Let {N,M} be the number of feature points of the input image pair $\left\{I_{a},I_{b}\right\}$ after ORB+BF, respectively. $X=\{x_{1},x_{2},...,x_{n}\}$ is all the feature matching neighborhoods from image $I_{a}$ to $I_{b}$. *X* can be classified as true or false by measuring the local support of each matching pair through GMS. $X_{i}\subseteq X$ is the subset of all neighborhoods. $S_{i}$ is the neighborhood support, which is expressed as follows:(1)$$S_{i}=\left|X_{i}\right|-1,$$
where −1 removes the original match from the sum.

GMS assumes that if the motion is smooth in a region, the correct matches have the same spatial locations across multiple neighborhood pairs, while false matches have relative spatial locations across multiple neighborhood pairs different. Equation (Equation 1) can be rewritten as follows:(2)$$S_{i}=\sum_{k=1}^{K}\left|X_{a^{k}b^{k}}\right|-1,$$
where K is the number of small neighborhoods predicted to move along with feature matching. $\left\{a^{k},b^{k}\right\}$ is the predicted region pair, $X_{a^{k}b^{k}}$ belonging to *X* is the feature matching subset.

Let $T^{ab}$ and $F^{ab}$ be the same and different positions of the region {a,b}, respectively. $f_{a}^{b}$ is the nearest neighbor feature in region *b* to one of the features in region *a*. $p_{t}=p\left(f_{a}^{b}|T^{ab}\right)$ and $p_{f}=p\left(f_{a}^{b}|F^{ab}\right)$. Assuming that each region pair {a,b} has {n,m} feature points, then:(3)$$\left\{\begin{array}{c} p_{t}=t+\left(1-t\right)\beta m/M\\ p_{f}=\beta \left(1-t\right)\left(m/M\right) \end{array}\right.,$$
where β is the adjustment factor. Thus, $S_{i}$ be the number of matches in the neighborhood of $x_{i}$, the distribution of it follows the binomial distribution in Equation (Equation 2):(4)$$S_{i}∼\left\{\begin{array}{cc} B(Kn,p_{t}), & \mathrm{if}\;x_{i}\;\mathrm{is}\;\mathrm{ture}\\ B(Kn,p_{f}), & \mathrm{if}\;x_{i}\;\mathrm{is}\;\mathrm{false} \end{array}\right..$$
where *K* is the number of disjoint regions which match *i* predicts move together.

To calculate $S_{i}$ more efficiently, the image is divided into 20×20 grids as in [25]. The score $S_{ij}$ for the cell pair {i,j} is calculated as follows:(5)$$S_{ij}=\sum_{k=1}^{K=9}\left|X_{i^{k}j^{k}}\right|,$$
where $|X_{i^{k}j^{k}}|$ is the number of feature matches in the nine-square grid. The score threshold τ is used to distinguish whether the feature matching is correct. If $S_{ij}>\tau$, the matching $x_{i}$ at grid {i,j} is true; otherwise, $x_{i}$ is false.

In this way, we obtain a denser set of feature correspondences compared with the traditional SIFT+RANSAC pattern. It will be used as input for the subsequent warping.

### 3.2. Hybrid Terms Optimization Warp

#### 3.2.1. Mathematical Preparation

Let $X_{1}={\left[x_{1},y_{1},1\right]}^{T}$ and $X_{2}={\left[x_{2},y_{2},1\right]}^{T}$ denote the feature points obtained by the feature correspondences increase in the input images $I_{a}$ and $I_{b}$, respectively. The alignment of the overlapping region of the two images is a linear transformation from $I_{a}$ to $I_{b}$ in homogeneous coordinates, which is defined below:(6)$$X_{2}=HX_{1},$$
where *H* is defined as a 3×3 matrix. If it is a global homography transformation, then we have the following:(7)$$H_{gh}=\left[\begin{array}{ccc} h_{1} & h_{2} & h_{3}\\ h_{4} & h_{5} & h_{6}\\ h_{7} & h_{8} & h_{9} \end{array}\right].$$
Similarly, if it is a global similarity transformation, we have the following:(8)$$H_{gs}=\left[\begin{array}{ccc} s\cos\theta & s\sin\theta & t_{x}\\ s\sin\theta & s\cos\theta & t_{y}\\ 0 & 0 & 1 \end{array}\right],$$
where *s* and θ are the scale and angle of the transformation, respectively. $t_{x}$ and $t_{y}$ are the translation distances along the x and y directions, respectively.

In inhomogeneous coordinates:(9)$$x_{2}^{\prime }=\frac{h_{1}x_{1}+h_{2}y_{1}+h_{3}}{h_{7}x_{1}+h_{8}y_{1}+h_{9}},\;y_{2}^{\prime }=\frac{h_{4}x_{1}+h_{5}y_{1}+h_{6}}{h_{7}x_{1}+h_{8}y_{1}+h_{9}}.$$
Next, we would like to solve the homography $H_{gh}$ and $H_{gs}$. Even though these inhomogeneous equations involve the coordinates non-linearly, the coefficients of *H* appear linearly. Given N pairs of feature correspondences, we can form the following linear system of equations:(10)Ah=0,
where $A\in R^{2N\times 9}$. It is obtained by transforming Equation (Equation 9). Equation (Equation 10) can be solved using homogeneous linear least square methods like Singular Value Decomposition (SVD). From SVD, the right singular vector that corresponds to the smallest singular value is the solution h, and then we reshape h into the matrix $H_{gh}$. For similarity transformation, we filter out the similarity matrix corresponding to the rotation angle with the smallest angle according to the feature points as the global similarity $H_{gs}$.

Global homography has difficulty producing an accurate alignment when the scenes are not coplanar. In contrast, APAP uses spatially varying warping $H_{*}$, which expands the homography *H* into multiple local homographies applied to each grid cell, and it uses moving DLT for image alignment. This allows for better alignment in the overlapping region. Similar to the method in [42], assuming that the feature point *p* in the image $I_{a}$ can be represented by a linear combination of a vector composed of four vertices of the mesh $V={\left[v_{1},v_{2},v_{3},v_{4}\right]}^{T}$. It can be expressed as: p=WV, and the bilinear interpolation weight of the four mesh vertices $W={\left[w_{1},w_{2},w_{3},w_{4}\right]}^{T}$. Then, these mesh vertices are formed into the transformed mesh vertices $\hat{V}={\left[\hat{v_{1}},\hat{v_{2}},\hat{v_{3}},\hat{v_{4}}\right]}^{T}$ after mesh deformation. Therefore, the deformed vertex coordinate $\hat{p}$ is calculated as the bilinear interpolation as follows:(11)$$\hat{p}=w_{1}v_{1}+w_{2}v_{2}+w_{3}v_{3}+w_{4}v_{4}.$$
As a result, we convert point correspondence constraints to mesh vertex correspondence constraints.

After the above preparation, the energy function can be defined as:(12)$$E=E_{a}+E_{d}+E_{s},$$
where $E_{a}$ addresses the alignment issue by feature correspondences increase model to increase the number of correct matching (see Section 3.2.2), $E_{d}$ addresses the distortion issue by cross lines with a novel hybrid initial warp prior (see Section 3.2.3), and $E_{s}$ addresses the salient content-preserving issue by protecting both local and global lines from being bent in the non-overlapping region (see Section 3.2.4). The mesh warping result is obtained by minimizing *E* (see Section 3.2.5). As all constraint terms are quadratic, they can be solved and minimized by any sparse linear solver.

#### 3.2.2. Alignment Term

The point alignment item $E_{p}$ is used to align the two images $\left\{I_{a},I_{b}\right\}$, which is defined as follows:(13)$$E_{p}=\sum_{i}||W_{i}\hat{V_{i}}-p_{i}^{\prime }|{|}^{2}=\left|\right|\hat{p}-p^{\prime }{\left|\right|}^{2},$$
where $p^{\prime }$ indicates the matching points in the image $I_{b}$. Line alignment is also considered as an alignment supplement. Given a set of line correspondences $\left\{l_{aj},l_{bj}\right\}$, where $l_{aj}\in I_{a}$ is represented by the line segment with the endpoints $P_{ak}$, and $P_{ak}=W_{pk}\hat{V_{pk}}$. $l_{bj}\in I_{b}$ is represented by the line equation $a_{j}x+b_{j}y+c_{j}=0$ [18], then the line alignment term is represented as below:(14)$$E_{l}=\sum_{j,k}||l_{bj}^{\prime T}P_{ak}\lambda_{j}|{|}^{2},$$
where $\lambda_{j}=\frac{1}{\sqrt{a_{j}^{2}+b_{j}^{2}}}$. With the scalar $\lambda_{j}$, the weight between two endpoints is balanced in terms of geometric meaning.

In summary, the alignment term is represented as:(15)$$E_{a}=\lambda_{p}E_{p}+\lambda_{l}E_{l}.$$

#### 3.2.3. Distortion Term

Both global homography and local homography can cause distortion in the non-overlapping region of the image. This distortion is negligible for content close to the common main plane, but more severe for content further away from the main plane. In order to control the distortion of the target image $I_{a}$, a series of horizontal and vertical lines are constructed, called cross lines. They are regarded as the intrinsic structure of the image and are used to reflect the overall warping degree of the target image $I_{a}$.

In [40], s(x,y,k) is the slope of the line in $I_{b}$ corresponding to the line passing (x,y) with slope *k* in $I_{a}$. $l_{u}$ is the warped horizontal line and $l_{v}$ is the vertical line that is closest to the boundary between overlapping and non-overlapping areas. If a homography warp is given, there is a unique set of parallel lines corresponding to it. The slopes of the parallel lines before and after warping are as follows:(16)$$k_{1}=-\frac{h_{7}}{h_{8}},\;s\left(x,y,k1\right)=\frac{h_{4}h_{8}-h_{5}h_{7}}{h_{1}h_{8}-h_{2}h_{7}}.$$
Therefore, $l_{v}$ can be set to be the split line with slope $k_{1}$ that is closest to the boundary of the overlapping region and non-overlapping region, and $l_{u}$ can be set to be orthogonal to $l_{v}$ before and after warping, as shown below:(17)$$k_{1}\cdot k_{2}=-1,\;s\left(x,y,k1\right)\cdot s\left(x,y,k2\right)=-1,$$

Although cross lines can effectively alleviate the distortion of the global homography and balance the projection and perspective distortion in the non-overlapping region, it can produce a severely stretched single-perspective stitching result when the scene depth or parallax is large (see Figure 4b). Furthermore, it relies on the initial global homography. If the global homography transformation is misaligned in the overlapping region and severely distorted in the non-overlapping region, the cross lines can only alleviate less distortions and cannot flexibly deal with the distortion and misalignment problems without a fundamental solution that the distortion in the non-overlapping region caused by homography. In [17], the global similarity transformation can effectively mitigate the distortion in the non-overlapping region by iteratively finding the minimum rotation angle of the transformation. Therefore, we mix the global homography transformation with the global similarity transformation, and we adjust the rotation angle of the cross lines by adding weights, so as to reduce the distortion flexibly (as shown in Figure 4d).

In this paper, we change the initial global homography to a hybrid global homography transformation and a global similarity transformation as shown in Figure 5. This is defined as follows:(18)$$H_{init}=H_{gh}+H_{hy},$$
where $H_{init}$, $H_{gh}$, $H_{hy}$ are the initial transformation, the global homography transformation, and the hybrid adjustment terms, respectively. They are all 3×3 matrices. The purpose is to accurately align in the overlapping region, reduce distortion in the non-overlapping region, and flexibly cope with large parallax and large scene depth. The hybrid adjustment term is defined as follows:(19)$$H_{hy}=W_{hy}\left(H_{gs}-H_{gh}\right),$$
where $H_{gs}$ is the global similarity transformation, and $W_{hy}$ is the weighting factor in the range of [0,1) for adjusting $H_{gs}$ and $H_{gh}$. If $W_{hy}$ is 0, $H_{init}$ is the global homography transformation $H_{gh}$, at this time the slope of the cross line in Equation (Equation 15) is the largest. Since the overall warping change is the largest, the distortion is the most serious. However, the alignment is better; if $W_{hy}$ is 1, $H_{init}$ is the global similarity transformation $H_{gs}$, but the slope of the cross lines does not exist. The closer the value of $W_{hy}$ is to 0, the higher the slope; the higher the degree of warp, the better the alignment, but the more serious the distortion. On the contrary, the closer the value of $W_{hy}$ to 1, the smaller the slope, the smaller the transformation angle, the less distortion, but the more serious the misalignment. Therefore, it is necessary to choose the appropriate value of $W_{hy}$ according to different types of images to be stitched. The stitching results at different $W_{hy}$ are shown in Figure 4.

By the method discussed above, we divide the distortion term $E_{d}$ into a perspective distortion term $E_{ps}$ that preserves the perspective given by the new hybrid homography warp in Equation (Equation 18) and a projection distortion term $E_{pj}$ that mitigates the projective distortion.

Given the set of cross line correspondences $\left\{l_{ai}^{u},l_{bi}^{u}\right\}$ and $\left\{l_{aj}^{v},l_{bj}^{v}\right\}$ that are parallel to $l_{u}$ and $l_{v}$, the points of cross lines are recorded as $\left\{p_{k}^{ui}\right\}$ and $\left\{p_{k}^{vi}\right\}$, then
(20)$$\begin{array}{cc} E_{ps} & =\sum_{i=1}\sum_{k=1}{\left|\langle W_{p_{k+1}^{ui}}{\hat{V}}_{p_{k+1}^{ui}}-W_{p_{k}^{ui}}{\hat{V}}_{p_{k}^{ui}},{\vec{n}}_{i}^{u}\rangle \right|}^{2}\\ \; & +\sum_{j=1}\sum_{k=1}{\left|\langle W_{p_{k+1}^{vj}}{\hat{V}}_{p_{k+1}^{vj}}-W_{p_{k}^{vj}}{\hat{V}}_{p_{k}^{vj}},{\vec{n}}_{j}^{v}\rangle \right|}^{2}\\ \; & +\sum_{j=1}\sum_{k=1}{∥W_{p_{k}^{vj}}{\hat{V}}_{p_{k}^{vj}}+W_{p_{k+2}^{vj}}{\hat{V}}_{p_{k+2}^{vj}}-2W_{p_{k+1}^{vj}}{\hat{V}}_{p_{k+1}^{vj}}∥}^{2}, \end{array}$$
where ${\vec{n}}_{i}^{u}$ and ${\vec{n}}_{j}^{v}$ are the normal vectors of $l_{bi}^{u}$ and $l_{bj}^{v}$. In Equation (Equation 19), the first two terms preserve the slopes of $l_{bi}^{u}$ and $l_{bj}^{v}$, and the last term preserve the ratios of lengths on $l_{aj}^{v}$.

The projection distortion term for the non-overlapping region of image $I_{a}$ is defined as follows:(21)$$E_{pj}=\sum_{i=1}\sum_{k=1}{∥W_{p_{k}^{ui}}{\hat{V}}_{p_{k}^{ui}}+W_{p_{k+2}^{ui}}{\hat{V}}_{p_{k+2}^{ui}}-2W_{p_{k+1}^{ui}}{\hat{V}}_{p_{k+1}^{ui}}∥}^{2}.$$
In fact, Equation (Equation 20) linearizes the scale on $l_{ai}^{u}$ in the non-overlapping region of $I_{a}$.

In summary, the distortion term is represented as:(22)$$E_{d}=\lambda_{ps}E_{ps}+\lambda_{pj}E_{pj}.$$

#### 3.2.4. Salient Term

The salient term is used to preserve the line structure in the non-overlapping region of $I_{a}$. Given the set of salient lines $\left\{l_{ak}^{s}\right\}$, each $l_{ak}^{s}\in I_{a}$ is denoted by the set of endpoints $p_{j}^{k}$, then:(23)$$E_{ls}=\sum_{k=1}\sum_{j=1}{∥\langle W_{p_{j+1}^{k}}{\hat{V}}_{p_{j+1}^{k}}-W_{p_{j}^{k}}{\hat{V}}_{p_{j}^{k}},{\vec{n}}_{k}\rangle ∥}^{2},$$
where ${\vec{n}}_{k}$ is the normal vector of $\left\{l_{bk}^{s}\right\}$ that is calculated from the new homography $H_{init}$ in Equation (Equation 17).

Due to the limitations of line segment detection, the detected line segment is generally not very long, and it may not be the actual line structure affected by the viewing direction or other factors. When some local lines are actually collinear, if they are not globally optimized, there will be significant global line bending after warping. Inspired by [20], we merge some apparently collinear local line segments into global lines, while adding the global line term $E_{gs}$ to the salient term.

Given the set of global lines $\left\{l_{ag}^{s}\right\}$, each $l_{ag}^{s}\in I_{a}$ is denoted by the set of endpoints $p_{k}^{g}$, then we have the following:(24)$$E_{gs}=\sum_{g=1}\sum_{k=1}{∥\langle W_{p_{k+1}^{g}}{\hat{V}}_{p_{k+1}^{g}}-W_{p_{k}^{g}}{\hat{V}}_{p_{k}^{g}},{\vec{n}}_{g}\rangle ∥}^{2},$$
where ${\vec{n}}_{g}$ is the normal vector of $\left\{l_{bg}^{s}\right\}$. Figure 6 also shows the effect of adding the global line term on the stitching results.

Similarly, local lines term and global line term can be represented as follows:(25)$$E_{s}=\lambda_{ls}E_{ls}+\lambda_{gs}E_{gs}.$$

#### 3.2.5. Total Energy Function

The above terms can be written as a total energy function:(26)$$\begin{array}{c} E=\lambda_{p}E_{p}+\lambda_{l}E_{l}+\lambda_{ps}E_{ps}+\lambda_{pj}E_{pj}+\lambda_{ls}E_{ls}+\lambda_{gs}E_{gs}. \end{array}$$The above function is quadratic, and this turns into an optimization problem which can be solved and minimized by any sparse linear solver.

To sum up, we define alignment terms, distortion terms, and salient terms based on a novel hybrid initial warp prior to address alignment and distortion issues. Among them, the point alignment term in the alignment term is based on our proposed feature correspondences increase model. The stitching process of the two images is shown in Algorithm 1.
**Algorithm 1** Stitching two images.**Input:** a target image Ia and a reference image Ib.**Output:** a stitched image.
  1:Match point and line features between Ia and Ib to obtain {pa,pb} and {la,lb} via feature correspondences increase.  2:Calculate a global homography Hgh and a global similarity matrix Hgs via dual-feature.  3:Calculate a hybrid warp Hinit by Equation (Equation 18) and Equation (Equation 19).  4:Calculate {laiu,lbiu} and {lajv,lbjv} from Hinit by Equation (Equation 16) and Equation (Equation 17).  5:Detecting salient line segments in the non-overlapping region to obtain {laks}.  6:Merge collinear local line segments into global lines {lags}.  7:Uniformly sample {laiu},{laiv},{laks},{lags}.  8:Solve V^ via minimizing the total energy term in Equation (Equation 26).  9:Warp Ia via the bilinear interpolation on the basis of V^.  10:Stitching the warped Ia with Ib via linear blending or seam-cutting blending.


## 4. Experiments

### 4.1. Experimental Setup

The proposed method has been tested on selected datasets, including some public datasets with 10 pairs of images in [8,10,17,20,34,39], and our own collected datasets with 5 pairs of images. Some input image pairs as examples are shown in Figure 7 for quantitative evaluation and qualitative comparison. All the images are natural RGB ones. In addition to regular natural images, we focus on images with low-textured or repetitively-textured regions in the overlapping region, which will better demonstrate the effectiveness of the proposed method.

We compare our approach with some state-of-the-art methods, including homography, APAP, AANAP, SPHP, SPW and LPC. The parameters of the existing methods are set as suggested by the original papers. In order to facilitate the comparison of alignment effects in the overlapping region and structure preservation effects in the non-overlapping region, while avoiding seam-cutting blending to mask the original misalignment, all the methods use linear blending. In addition, we also show the visual enhancement brought by seam-cutting blending in [37].

In the experiments, we first extract and match ORB feature points, then we remove mismatches with GMS, while using LSD [24] to detect line segments and match them by [21]. The number of ORB feature points is 30,000. Since the image stitching task does not involve rotation and scaling, the parameters of rotation and scaling in GMS are both 0. For the hybrid terms optimization parameters, the deformation mesh size is set to 40 × 40, and the inputs $\lambda_{p}$, $\lambda_{l}$, $\lambda_{ps}$, $\lambda_{pj}$, $\lambda_{ls}$, and $\lambda_{gs}$ are set to 1, 5, 50, 100, 50, and 100, respectively. The initial homography fine-tuning weight $W_{hy}$ is set to 0.4. It is worth noting that $W_{hy}$ and $\lambda_{ps}$ are inversely proportional. If $W_{hy}$ is too large and the transformation is approximately similar to the global one, the perspective relationship will change significantly. Therefore, the perspective term should be reduced to ensure alignment of the overlapping regions. In this way, the non-overlapping region is close to the rectangle with the least distortion, and the distortion term and the salient term can be appropriately reduced, and vice versa. Since the salient term and the distortion term are competing in [22], the ration of the inputs $\lambda_{pj}$ and $\lambda_{gs}$ are 1:1.

The proposed model is mainly implemented in MATLAB (some portions of the code are written in C++), and all the experiments are carried out on a PC with AMD Ryzen 7 4800 H 2.90 GHz CPU and 16 GB RAM.

### 4.2. Quantitative Evaluation

We quantitatively evaluate the stitching quality of different single-perspective methods using SSIM [43] in the range of [0, 1]. The larger the value of SSIM, the more similar the structure after warping. The overlapping regions of each pair of images in the dataset are extracted to compute the corresponding SSIM after alignment.

Besides, we quantitatively evaluate the alignment accuracy of the proposed method by root mean squared error (RMSE) on feature correspondences:(27)$$RMSE\left(f\right)=\sqrt{\frac{1}{N}\sum_{i=1}^{N}{∥f\left(p_{i}\right)-p_{i}^{\prime }∥}^{2}},$$
where *N* is the number of feature correspondences and f represents a planar warp. Table 1 shows the number of matches, SSIM and RMSE using different methods on different datasets. Among them, the first four rows are dense texture images with small parallax, the fifth to eighth rows are images with large parallax and no obvious low-textured areas, and the last seven rows are images with obvious low-textured areas in the overlapping region. The traditional SIFT+RANSAC model is used as a baseline model for comparison. Compared to it, our feature correspondences increase model significantly increases the number of matches generally. The more feature correspondences there are, the greater the image information entropy is, and the better the image detail performance is, so as to obtain a better stitching result. In particular, it allows sufficient matches for grid-based stitching algorithms, especially for low-textured images. Additionally, the values of SSIM and RMSE also show that our method outperforms other methods, especially that our method achieves the highest scores on images with significantly weakly textured or repetitively textured regions. On other types of images, although some scores are low, the scores of our method are comparable with other methods.

### 4.3. Qualitative Comparison

First of all, we replace the traditional SIFT/SURF + RANSAC pattern in each method with the proposed feature correspondences increase model. Figure 8 compares the stitching results of some methods before and after applying feature correspondences increase on the dataset *intersection* [17]. As in the original three methods shown in Figure 8a, the inliers are mostly concentrated in the house area as indicated by the blue circle, so the alignment around the house area is better. Since the feature points are only concentrated in the dense-textured region, the feature correspondences of low-textured region is insufficient, which leads to serious misalignment, as shown in the red rectangle. APAP is a grid-based local alignment method, and some pixels will destroy the continuity and consistency of the original texture. However, SPW, LPC and SPHP do not destroy the original texture of the image due to the restriction of alignment terms. Figure 8b shows the results after applying our proposed method. Due to the increase of feature correspondences, the alignment of the three methods is more accurate in low-textured regions. The local alignment property of APAP itself causes it to suffer structural damage, but with less distortion in the non-overlapping region. It also illustrates that grid-based local alignment relies on a sufficient and efficient distribution of the number of feature points. In the case of increased feature correspondences, SPW and LPC ignore the distortion of non-overlapping regions due to over-focusing on alignment. It results in a less smooth transition between overlapping and non-overlapping regions. LPC is better than SPW, but it still has artifacts. SPHP has more accurate alignment compared to using SIFT+RANSAC. However, due to its excellent distortion preserving properties, alignment is not as accurate as SPW and LPC. The above experiments indicate that our proposed feature correspondences increase model can be added to any feature point-based method to replace the original SIFT/SURF+RANSAC pattern to improve alignment accuracy. However, accurate alignment is often accompanied by inevitable distortions in the non-overlapping region. The next issue we want to focus on is how to balance alignment and distortion.

Figure 9 shows the stitching results of different methods on the dataset *runway*. The first column shows the stitching results. The runway areas in the red rectangles represent low-textured regions, and the magnified versions are shown in the second column. Similarly, the buildings in the blue rectangles represent the densely-textured regions, and their magnified versions are shown in the third column. Global homography does not yield satisfactory results due to the presence of parallax. There is obvious misalignment, either in the building or the runway. The results of APAP based on local alignment are slightly better than those of global alignment, but the feature points are concentrated in the densely-textured region, while they are insufficient in the low-textured regions. This leads to more accurate alignment in the building and serious misalignment in the runway. Further, global homography and APAP have the same projection and perspective distortion in the non-overlapping regions. The third and fourth rows demonstrate two methods of warping two images. While they effectively mitigate the distortion in the non-overlapping regions, insufficient inliers cause misalignment in low-textured regions, as shown in the red rectangles. The fifth row shows the results of TFA. Although it uses a two-step RANSAC outlier removal method to ensure the matching points as correct as possible, the characteristic of SIFT+RANSAC pattern leads to a serious lack of correct matches in the low-textured regions, which in turn leads to misalignment as shown in the red rectangles. Meanwhile, the triangulation method destroys the original structure, causing the line structure to be bent in the overlapping regions. In the non-overlapping region, its distortion preservation is better than global homography and APAP. In addition, SPW also shows misalignment in the overlapping regions. Although LPC aligned the densely-textured regions, it still caused serious misalignment of the runway areas due to insufficient feature points. Due to its more accurate alignment in the densely-textured regions in the blue rectangles, the distortion in the non-overlapping regions is more severe than SPW. To better demonstrate the effectiveness of our method, we provide three comparisons of results. The eighth row presents the results using only our feature correspondences increase (FCI) model. The ninth row shows the result of using only our hybrid terms optimization (HTO) warp but SIFT+RANSAC as the feature matching stage. The last row is the results of using both FCI and HTO. Using only FCI or HTO may not achieve the results we expect. Alignment using only FCI is significantly more accurate, but there is still slight distortion. Distortion is significantly alleviated by using only HTO but SIFT+RANSAC in the non-overlapping regions, and the target image is more rectangular, but the alignment of overlapping regions is less accurate. When both FCI and HTO are used, our method is aligned in the densely-textured regions. In addition, our alignment is very accurate in the low-textured regions compared to other methods. Furthermore, our method also outperforms SPW and LPC in terms of distortion in the non-overlapping regions. Our method achieves a balance between alignment and distortion to a certain extent.

### 4.4. Seam-Cutting Blending

The proposed method has achieved more accurate alignment and it effectively balances alignment and distortion by linear blending. Nonetheless, in some challenging large parallax scenes, our method may produce misalignment if only linear blending is used. Therefore, we introduce seam-cutting blending, which can effectively eliminate the artifacts and misalignment brought by linear blending.

The approach proposed in [10] is a typical seam-driven stitching method. Similar to the other methods we compared, it uses SIFT+RANSAC as feature matching, and it has similar optimization constraints. However, there is no line structure preservation term in its energy function. Figure 10 presents the comparison of [10] and our method on the dataset *025*. The red circles highlight the differences between the two methods in the overlapping regions. The results of [10] present significant bending of the line structures at the transparent glass building. In contrast, our method outperforms this approach in terms of line structures preservation since our salient term can effectively protect the line structures from being bent. Besides, our method is also slightly better for distortion in the non-overlapping regions.

In addition to applying seam-cutting blending to handle large parallax scenes to a certain extent, our method can also deal with continuous structure damage and heavy occlusion issues due to a more accurate alignment. In fact, seam-cutting blending also depends on the pre-alignment. If the pre-alignment is severely misaligned, seam-cutting blending may destroy the original texture structure, resulting in inconsistent and discontinuous stitching results due to the inaccurate pre-alignment. Figure 11 shows the difference between LPC and our proposed method using seam-cutting blending on a large parallax example in our own dataset. It can be seen that the linear blending results of our proposed method are better than those of LPC with a different optimal seam. As a result, the seam-cutting blending of LPC destroys the original line structures, resulting in an obvious misalignment. This is shown in the blue circle in Figure 11e. In Figure 11f, our method achieves accurate alignment.

Likewise, if the object has a serious occlusion relationship between the foreground and the background, the degree of post-warp alignment will seriously affect the final results. Figure 12 shows an example with a front-to-back occlusion relationship via [19]. It can be seen that the optimal seam of our method effectively avoids the occlusion of the target foreground and background, and achieves accurate alignment (see Figure 12d,f).

## 5. Conclusions

In this paper, we propose a novel feature correspondences increase and hybrid terms optimization warp for image stitching. It can deal with the misalignment issue caused by insufficient feature correspondences of low-textured areas in the overlapping region and the distortion issue in non-overlapping regions, with better balancing alignment and distortion. The proposed method with seam-cutting blending can solve the issue of large parallax and occlusion. Both quantitative evaluation and qualitative comparison experiments show that the proposed method is more accurate in alignment and less distortion in the non-overlapping region compared with other methods, especially on images with low-textured areas in the overlapping region.

## Figures and Tables

**Figure 1 entropy-25-00106-f001:**
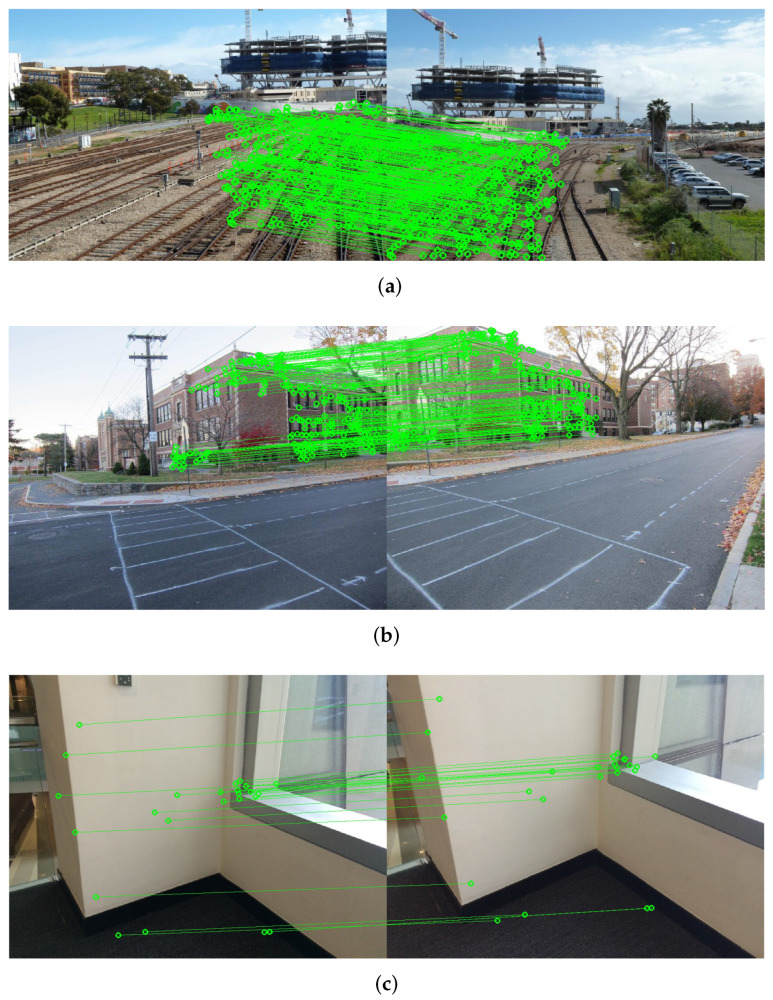
Comparing the number and distribution of inliers on different texture images. (**a**) Dense texture. (**b**) Semi-dense texture. (**c**) Low texture. The images are from [8,17,18], respectively.

**Figure 2 entropy-25-00106-f002:**
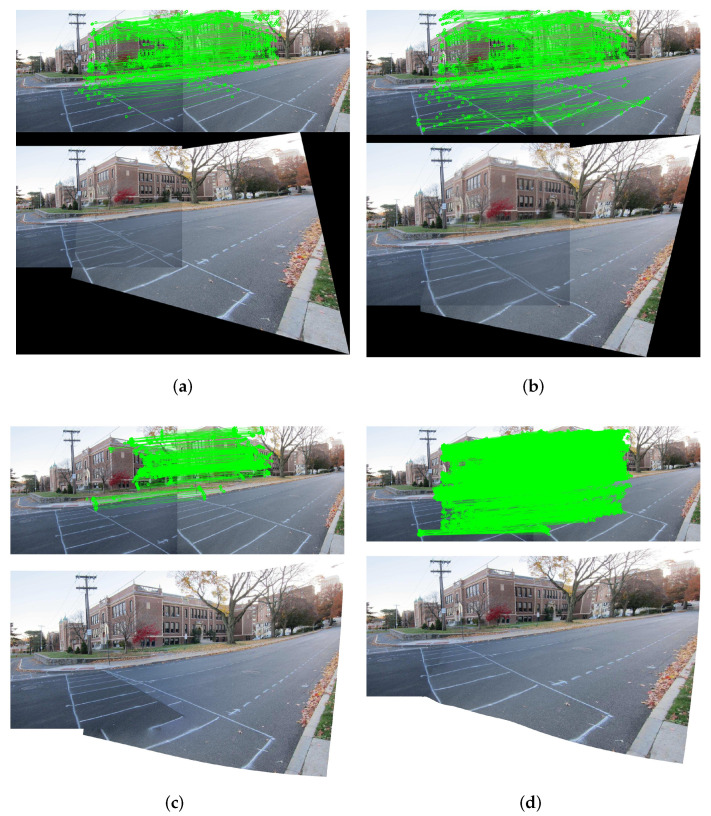
Comparison of inliers and stitching results of different methods. For each subfigure, the feature correspondences are on the top, and the stitching results are on the bottom. (**a**) APAP with RANSAC threshold 0.1. (**b**) APAP with RANSAC threshold 0.5. (**c**) LPC. (**d**) Ours.

**Figure 3 entropy-25-00106-f003:**
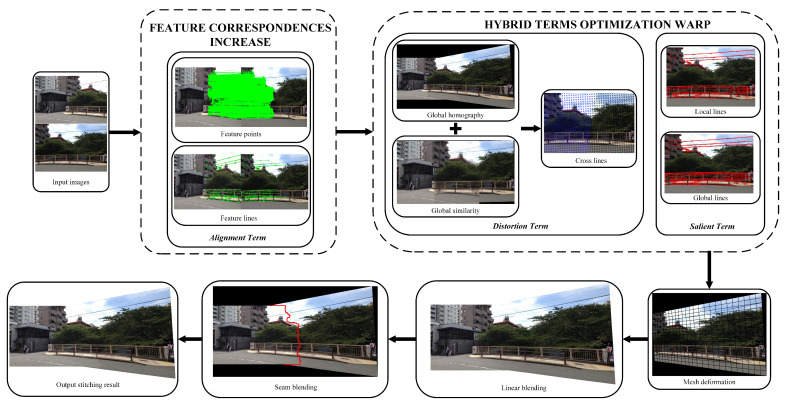
The overall process architecture of our method. The feature correspondences increase module mainly increases the number of feature points to ensure accurate alignment. The hybrid terms optimization warp is based on the above stage to obtain the optimal mesh deformation, which further balances the alignment and distortion.

**Figure 4 entropy-25-00106-f004:**
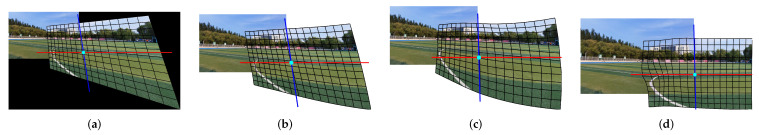
The cross lines distortion mitigation results. (**a**) Warped mesh result of homography. (**b**) Warped mesh result of SPW. (**c**) Warped mesh result of LPC. (**d**) Warped mesh result of ours. The blue line is closest to the border of the overlapping region and the non-overlapping region, and the red line is perpendicular to it. The smaller the slope of the cross lines, the less distortion.

**Figure 5 entropy-25-00106-f005:**
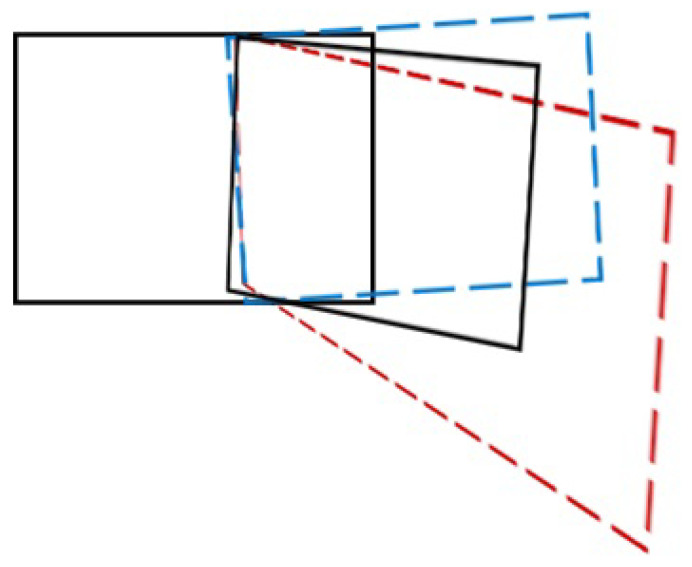
Hybrid global transformation. For the target image, the red dashed frame represents the global homography transformation and the blue dashed frame represents the global similarity transformation. The solid black frame in the middle represents the hybrid global transformation.

**Figure 6 entropy-25-00106-f006:**
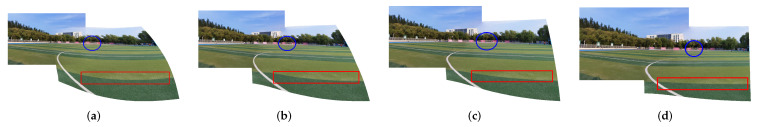
Global lines and hybrid transformation with different $W_{hy}$. (**a**) No global lines and $W_{hy}=0$. (**b**) With global lines and $W_{hy}=0$. (**c**) With global lines and $W_{hy}=0.4$. (**d**) With global lines and $W_{hy}=0.9$. The blue circles show the distortion mitigation effect with different $W_{hy}$, and the red rectangles show the global lines for structure preservation.

**Figure 7 entropy-25-00106-f007:**
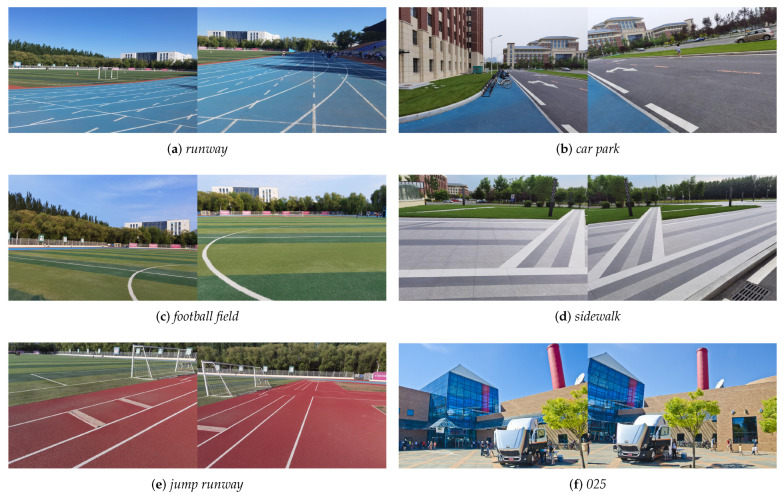
Datasets for experimental demonstration. (**a**–**e**) are our own images. (**f**) is from [10].

**Figure 8 entropy-25-00106-f008:**
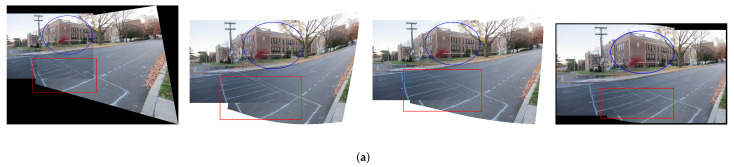
Comparison of the stitching results for increasing the number of features. (**a**) The original methods of APAP, SPW, LPC and SPHP, respectively. (**b**) After applying our method. The red boxes represent the low texture region, and the blue circles represent the dense texture region.

**Figure 9 entropy-25-00106-f009:**
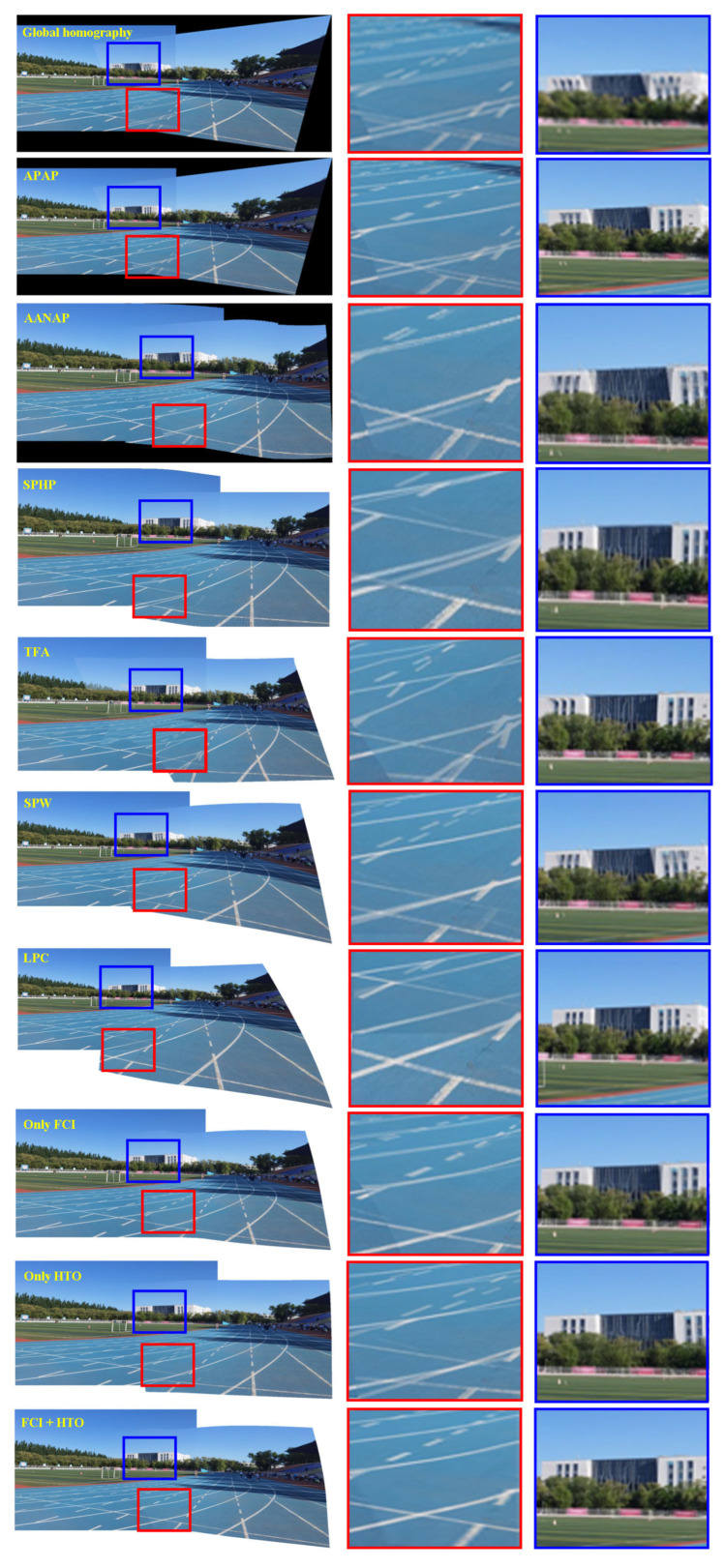
Comparison of stitching quality on *runway*. Row 1: Results using global homography. Row 2: Results using APAP. Row 3: Results of AANAP. Row 4: Results using SPHP. Row 5: Results of TFA. Row 6: Results of SPW. Row 7: Results of LPC. Row 8: Results only using our feature correspondences increase (FCI) model. Row 9: Results only using our hybrid terms optimization (HTO) warp but SIFT+RANSAC. Row 10: Results of our approach by using both FCI and HTO.

**Figure 10 entropy-25-00106-f010:**
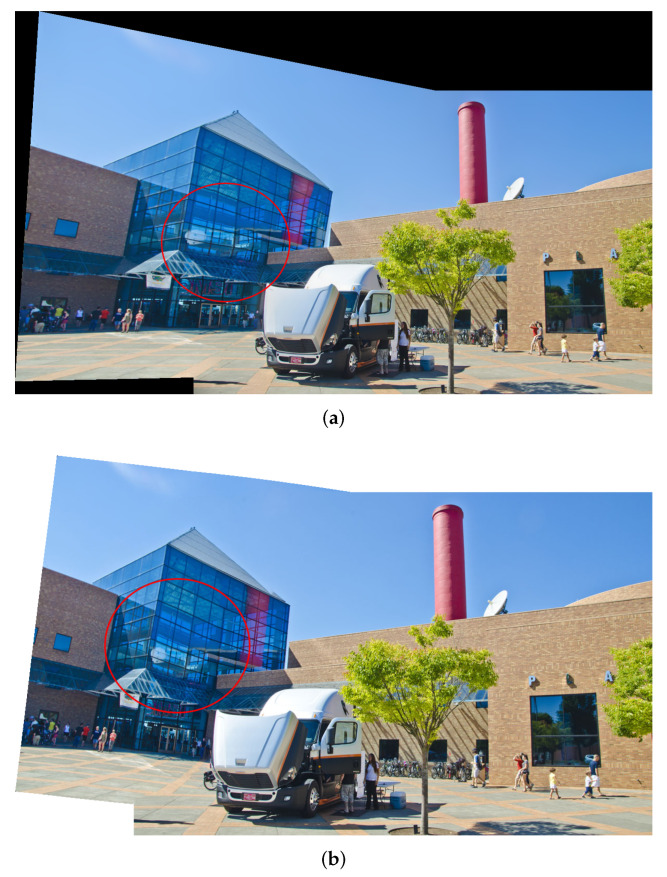
Comparison of [10] and our method on dataset *025*. (**a**) Stitching result of [10]. (**b**) Stitching result of our method. The red circles highlight the differences between the two methods in the overlapping regions. Our method protects the line structure better from being bent.

**Figure 11 entropy-25-00106-f011:**
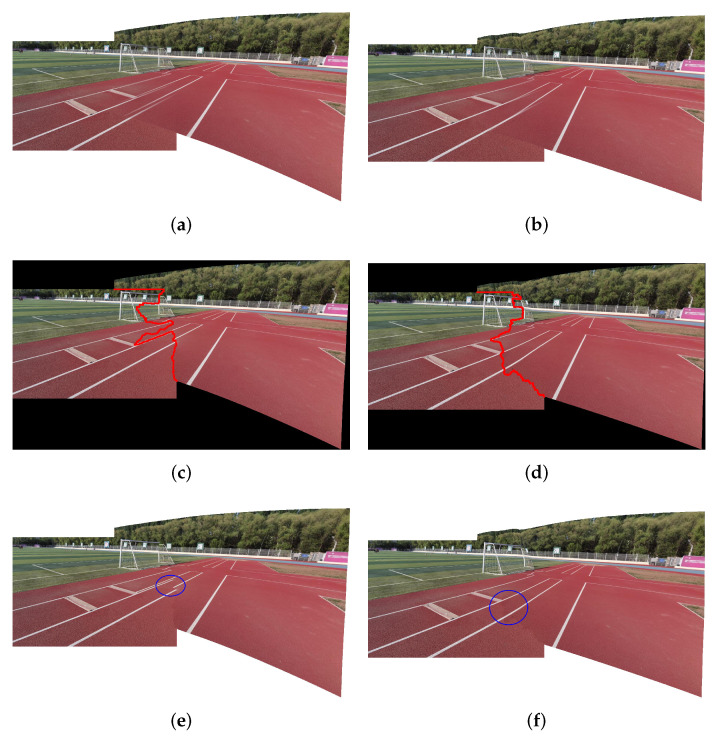
Comparison seam-cutting blending effects of LPC and ours. (**a**) Stitching result of LPC using linear blending. (**b**) Stitching result of ours using linear blending. (**c**) Optimal seam of LPC. (**d**) Optimal seam of ours. (**e**) Stitching result of LPC. (**f**) Stitching result of ours. The difference is shown in the blue circle. It is obvious to see that the seam blending of LPC destroys the original texture structure owing to misalignment seriously. In contrast, our method produces better seam-cutting blending due to more accurate alignment.

**Figure 12 entropy-25-00106-f012:**
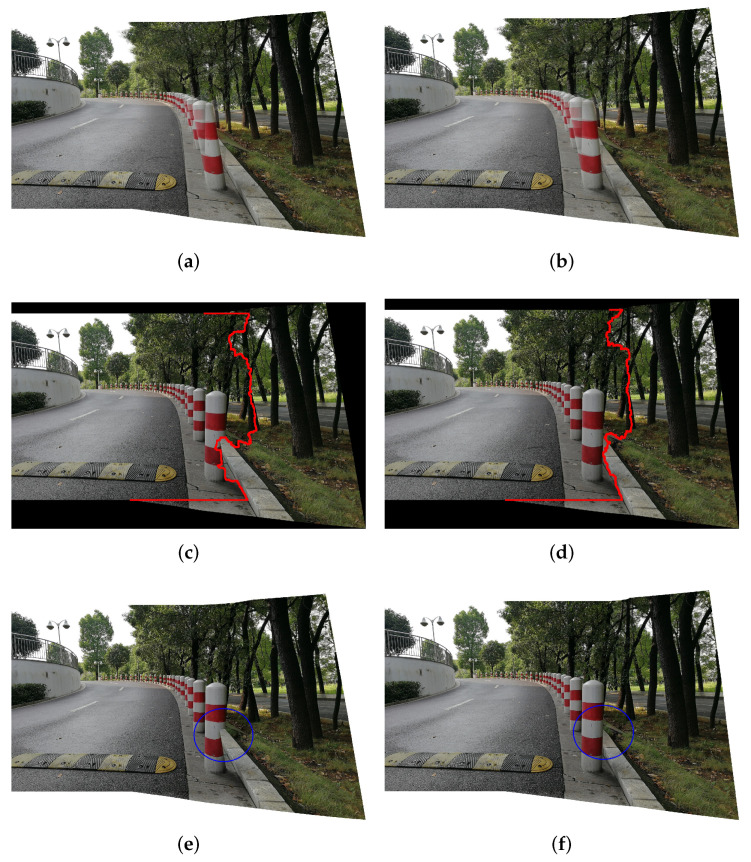
(**a**–**f**) Seam-cutting blending results of images with severe occlusions. The red line indicates the optimal seam of LPC and ours. The seam of the LPC destroys the relationship between the foreground and background of the object in the blue circle. However, our method is capable of aligning accurately enough to avoid this problem.

**Table 1 entropy-25-00106-t001:** Comparison on matches, SSIM and RMSE.

Databases	SIFT+RANSAC	Global Homography		APAP		SPW		LPC		OURS		
	Matches	RMSE	SSIM	RMSE	SSIM	RMSE	SSIM	RMSE	SSIM	Matches	SSIM	RMSE
*fence [17]*	597	0.6489	1.7674	0.7011	1.7773	0.6952	1.4592	0.6807	**1.3368**	5911	**0.7184**	1.5014
*Potberry [20]*	360	0.3971	2.3419	0.5904	2.3419	0.4648	2.0559	0.5144	1.8928	3772	**0.6142**	**1.7384**
*railtracks [8]*	651	0.4474	2.5763	0.6354	2.5767	0.6361	2.217	0.5871	**1.5155**	6176	**0.6738**	1.6670
*DHW-temple [34]*	322	0.5517	2.6633	0.6799	2.6437	0.6057	2.2674	0.4999	2.3199	5474	**0.6935**	**1.6570**
*MemorialHall [39]*	64	0.5370	2.5169	0.5590	2.5169	0.5018	2.2710	0.5217	**1.7471**	1603	**0.5417**	1.8711
*017 [10]*	330	0.5871	3.0506	0.6151	3.3019	0.6254	2.3501	0.6139	2.6153	5064	**0.6273**	**2.4313**
*cup [20]*	159	0.4747	2.6436	0.5509	2.1149	0.4778	2.4807	0.4778	2.6069	3041	**0.5730**	**2.0057**
*office [20]*	181	0.5415	3.5423	0.6582	3.5423	0.6150	3.0166	0.6124	2.8839	2991	**0.6904**	**1.9996**
*intersection [17]*	426	0.3612	3.4626	0.4809	3.5073	0.4152	2.5591	0.4285	2.8868	4314	**0.5520**	**1.9684**
*tower [19]*	652	0.5734	3.2967	0.7601	3.2989	0.7702	2.2089	0.8259	1.6391	6409	**0.8501**	**1.6145**
*runway*	208	0.5180	3.0860	0.5892	3.0865	0.5510	2.7476	0.5584	2.0154	3846	**0.6632**	**1.4540**
*car park*	293	0.4044	2.6544	0.4573	2.6544	0.4313	2.4622	0.2328	3.7501	3565	**0.4978**	**2.3074**
*football field*	237	0.4881	3.1110	0.5828	3.1110	0.4872	2.5211	0.4251	1.9414	3641	**0.5879**	**1.6641**
*sidewalk*	245	0.7261	2.3933	0.7681	2.3933	0.7270	1.9704	0.4740	2.9211	3130	**0.7953**	**1.9711**
*jump runway*	117	0.7299	1.8474	0.7457	1.8470	**0.7535**	**1.6250**	0.6628	1.9676	2347	0.7499	1.7163

## Data Availability

The data presented in this study are available on request from the corresponding author. The data are not publicly available due to privacy.

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
