# Peer review of "Feature Correspondences Increase and Hybrid Terms Optimization Warp for Image Stitching"

_entropy, 2023, doi:10.3390/e25010106_

Round 1
Reviewer 1 Report
The authors present a new quadratic energy function for image stiching. From my point-of-view, the manuscript is interesting and publication is recommended. There are only some minor comments that are not mandatory for publication. But the authors might wish to follow some of the comments.
o Abstract: The meaning of the sentence "Whether the matching...." is not clear to me.
o line 40. What is meant by "...relies too much on the energy function,..."?
o line 44: What is meant by "distortion preservation performance"?
o line 102: "..to increase feature correspondences..." would sond better to me.
o line 140: "...will may lead..."
o line 194 "TThe"
o page 7: Several variables and designations for the GMS are not fully explained. One might only understand this part with the referenced paper [25], where the same designations have been used.
o page 8: Similar to page 7, some details of mapping the feature ponts p and p-hat can only be understood after referecing the papet [42].
o line 296: There might be some sparse solvers that work more efficient for the problem at hand. One could be a bit more specific here. What sparse solvers did you use?
Reviewer 2 Report
The paper presents a multi-objective optimisation approach for image stitching, where overlap regions have low texture/match features.
The overall paper is well organised. The method description is okay. There are some repetitive descriptions in the introduction and 'related' work section. Please carefully check the acronyms - i.e. define them at their first instance in the paper. It is good to see the comparison in table 1.
It would be good if the authors could share the source code to allow readers to test the proposed method by using the readers' own image sets.
There are some spelling mistakes; please carefully check the script.
